**Data Availability Statement:** All relevant data are within the manuscript and its Supporting Information files.

# Evidence for key individual characteristics associated with outcomes following combined first-line interventions for knee osteoarthritis: A systematic review

Jacqui M. Couldrick[1]*, Andrew P. Woodward[1], M. Denika C. Silva[1], Joseph T. Lynch[1,2,3], Diana M. Perriman[1,2,3], Christian J. Barton[4], Jennie M. Scarvell[1]

**1** Faculty of Health, University of Canberra, Canberra, Australia, **2** College of Medicine and Health Sciences, Australian National University, Canberra, Australia, **3** Trauma Orthopaedic Research Unit, Canberra Hospital, Canberra, Australia, **4** La Trobe Sports and Exercise Medicine Research Centre, La Trobe University, Melbourne, Australia

* Jacqui.couldrick@canberra.edu.au

## Abstract

### Objective

To identify individual characteristics associated with outcomes following combined first-line interventions for knee osteoarthritis.

### Methods

MEDLINE, CINAHL, Scopus, Web of Science Core Collection and the Cochrane library were searched. Studies were included if they reported an association between baseline factors and change in pain or function following combined exercise therapy, osteoarthritis education, or weight management interventions for knee osteoarthritis. Risk of bias was assessed using Quality in Prognostic Factor Studies. Data was visualised and a narrative synthesis was conducted for key factors (age, sex, BMI, comorbidity, depression, and imaging severity).

### Results

32 studies were included. Being female compared to male was associated with 2–3 times the odds of a positive response. Older age was associated with reduced odds of a positive response. The effect size (less than 10% reduction) is unlikely to be clinically relevant. It was difficult to conclude whether BMI, comorbidity, depression and imaging severity were associated with pain and function outcomes following a combined first-line intervention for knee osteoarthritis. Low to very low certainty evidence was found for sex, BMI, depression, comorbidity and imaging severity and moderate certainty evidence for age. Varying study methods contributed to some difficulty in drawing clear conclusions.

**Funding:** The author(s) received no specific funding for this work.

## Conclusions

This systematic review found no clear evidence to suggest factors such as age, sex, BMI, OA severity and presence of depression or comorbidities are associated with the response to first-line interventions for knee OA. Current evidence indicates that some groups of people may respond equally to first-line interventions, such as those with or without comorbidities. First-line interventions consisting of exercise therapy, education, and weight loss for people with knee OA should be recommended irrespective of sex, age, obesity, comorbidity, depression and imaging findings.

## Introduction

Clinical practice guidelines recommend land-based exercise, education, and weight loss in those with knee osteoarthritis (OA) before undertaking total knee replacement [1]. The use of these first-line interventions remains suboptimal despite these recommendations [2]. Combined first-line interventions (or multi-component osteoarthritis interventions) consist of two or more non-surgical interventions of exercise therapy, osteoarthritis education and weight management [3]. Combined first-line interventions are increasingly provided internationally through specialist osteoarthritis management programs (OAMPs) as a complete package of care. These programs aim to deliver coordinated, evidence-based care to those with knee OA [3].

There is extensive research that demonstrates the effectiveness of exercise therapy for knee OA [4]. Less research has been conducted on outcomes following combined first-line interventions [5]. The evaluation of combined first-line interventions has been identified as a research priority, and recent reviews have examined their effectiveness [6, 7]. OAMPs such as the Good Life with osteoarthritis: Denmark (GLA:D®) have reported improvements in pain and function and a reduced desire for surgery [8]. However, a proportion of people undertaking these programs do not improve [8–10]. For example, data from the Swedish Better Management of Patients with Osteoarthritis (BOA) registry indicate that up to 43% of those with knee OA were considered responders based on NRS pain [9]. Immediate outcomes following the GLA:D® program indicates half of the participants were classified as a responder for pain and function outcomes [8]. Predicting those who may benefit from combined first-line interventions for knee OA is important. This may assist clinicians in the early identification of alternative treatments (such as pharmacological interventions), improve the timeliness or suitability for total joint replacement surgery, and assist medical practitioners in providing appropriate referrals to first-line care [2].

Research has begun to identify subgroups or individual characteristics associated with outcomes in those with knee OA following conservative treatments including intraarticular glucocorticoid injections [11] and combined first-line interventions [9, 12–16]. Many factors have been evaluated and include demographics, body mass index (BMI), comorbidities, psychological factors, and baseline disease severity [17, 18]. Interpreting the results of primary studies that have evaluated combined first line interventions is difficult due to a variety of factors identified, different study methods and contrasting results. A systematic review to collate these findings may provide clarity about which factors may influence the response to combined first-line interventions for knee OA and provide recommendations for clinical practice and future research.

This systematic review aimed to identify individual characteristics associated with a response to combined first-line interventions of land-based exercise therapy, OA education and weight loss for knee osteoarthritis. The primary objective of this systematic review was to identify baseline characteristics associated with improvements in pain and function following a combined first-line intervention in people with knee osteoarthritis. The secondary objective was to evaluate baseline characteristics associated with a change in the willingness to undertake surgery or undertake total knee replacement surgery.

## Methods

### Protocol and registration

This systematic review was registered (PROSPERO, protocol number CRD42021234398 (www.crd.york.ac.uk/prospero)). There were no amendments to the protocol. Reporting follows the PRISMA 2020 statement [19] (S1 Table).

### Eligibility criteria

Eligible studies investigated an association between a baseline prognostic factor and outcome following a multi-component intervention in those with knee osteoarthritis. Participants had knee osteoarthritis diagnosed either clinically or radiographically. The combined first-line intervention included (1) land-based exercise (any type) and either (2) arthritis education or self-management strategies or (3) weight loss or dietary management. The study design was not restricted and included secondary analysis of RCT data, case-series and cohort or longitudinal studies, including data from registries. The effect estimate was reported as a beta coefficient, odds ratio (OR), risk ratio (RR), hazard ratio (HR) or mean difference (MD). Primary outcome measures were a change in any measure of pain and function from baseline to follow-up with no restrictions to the length of follow-up. Secondary outcome measures were change in willingness to undertake joint replacement surgery or to have undertaken joint replacement.

Studies were excluded if participants had a TKR, had rheumatoid arthritis or other inflammatory conditions, studied pharmacological interventions, or did not estimate a prognostic factor at baseline with a reported association measure. Studies that examined treatment moderators or subgroup analysis that reported treatment effect sizes for baseline prognostic factors were excluded. Conference abstracts and review studies were excluded.

### Search strategy

MEDLINE and CINAHL (via EBSCO), Scopus, Web of Science Core Collection, and the Cochrane library were searched from inception to October 19th, 2022 (S2 Table). A health information specialist assisted in building the search strategy, which contained keywords related to knee osteoarthritis, multi-component non-surgical interventions and prognostic factors or measures of association. There were no limitations for year or language of publication. Electronic searches were complemented by manual searches of reference lists from included studies.

### Study selection process

Covidence systematic review software 2.0 (Melbourne, Australia, www.covidence.org) was used to manage the study selection process. Papers (title, abstracts, full texts) were independently screened by three investigators (JC, JL, and DS). A fourth reviewer (JS) resolved any conflicts at full-text screening.

## Data extraction

The modified version of CHARMS-PF checklist was used for data extraction [20]. Data extracted included study type, source of data, sample size and missing data, description of the intervention, outcomes to be predicted, the number and type of prognostic factors of interest, measurement of prognostic factors and cut-off points used, description of modelling method, reporting of adjusted or unadjusted effect estimates and the set of adjustment factors (covariates).

Data were extracted by the primary reviewer (JC) using Covidence and Microsoft Excel 2020. Secondary reviewers checked the strength of association measures and calculations (AW and JL).

## Quality assessment

Risk of bias was assessed using the Quality in Prognostic Factor Studies (QUIPS) tool [20]. QUIPS consists of six domains: study participation, study attrition, prognostic factor measurement, outcome measurement, adjustment for other prognostic factors, and statistical analysis and reporting. Each domain was rated as low, moderate, or high risk of bias. A study was deemed low risk of bias overall if all or most domains were rated as low. Suggested signalling items for each domain were discussed a priori with the research team. A study was rated low risk of bias if most signalling items had been addressed. If one signalling item was deemed quite problematic, this was weighted more heavily in the decision process.

Three independent reviewers (JC, JL, and DS) assessed risk of bias using Covidence software. Conflicts were resolved by discussion with all three reviewers.

## Data synthesis and analysis

The prognostic factors pre-identified in the study protocol were reported (age, sex, BMI, comorbidities, depression, and baseline OA severity). The effect estimate for each prognostic factor was summarised using odds ratio (OR), hazard ratio (HR), risk ratio (RR), or mean difference (MD) and the corresponding 95% confidence interval. Where relevant, both unadjusted and adjusted effect estimates were recorded. A meta-analysis was not performed due to the inconsistency in the intervention components, duration of follow-up, outcome measures, and methodology. Instead, a qualitative synthesis was conducted, and data visualisation was presented using R (https://www.r-project.org/) and 'ggplot2' (https://ggplot2.tidyverse.org/).

The term 'responder' was used to refer to whether a person had improved in either a pain or function outcome measure. This response may be a positive or a negative response to the treatment and will depend on the outcome measure used to evaluate the response.

For all observations to be on a common scale, the OR and 95% CI for the prognostic factors of age, sex, BMI, comorbidity, and depression were rescaled when necessary, such that an odds ratio greater than 1 represents a positive response. Many studies reported several related outcomes, which resulted in an OR being reported multiple times within the same study [10, 14]. Due to similarities in the outcome measures, we presented all multiple outcome measures within studies.

A recalculation for the prognostic factor of age was carried out, with the continuous cut-off point recalculated from age per 5 years to per year [15]. Two discrepancies observed in the reported data were resolved by contacting the author [10, 12]. Multiple outcome variables were reported in two studies [15, 21]. Gwynne-Jones (2018) used multinomial logistic regression with three categories, worse, stable and improved. OR for 'better' versus 'stable' was extracted instead of OR for 'worse' [15]. Lee (2018) used multinomial logistic regression and reported the OR for four pain and function trajectories measured over 12 weeks [21]. The

lower pain, early improvement trajectory (versus higher pain, no improvement) and the higher function, early improvement (versus lower function, delayed improvement) were extracted for comorbidities [21].

The certainty in the estimates of association were rated using the Grading of Recommendations Assessment, Development and Evaluation (GRADE) for prognostic factors [22, 23]. Each prognostic factor was rated from very low to high with consideration to the domains of risk of bias, inconsistency, indirectness, imprecision and publication bias [22].

## Results

### Study selection

Database searching identified 16 729 records. After duplicates were removed, 6931 records were screened, and 151 full-text articles were reviewed (Fig 1). 32 studies were included in the systematic review [10, 12, 14–18, 21, 24–48].

Reasons for exclusion were the intervention was defined retrospectively from electronic health records [49]; or retrospectively by participant self-report [50, 51], an exercise intervention only [52–56], including exercise and manual therapy [57] or the addition of joint injections with exercise and education component [58]. Other reasons included the study used analysis of variance and we were unable to extract meaningful estimates [59, 60], did not report an association measure [61] or used individual participant data (IPD) from 7 RCTs that evaluated a range of interventions and different musculoskeletal conditions, including knee osteoarthritis [62]. A study that examined pre-treatment pain sensitivity was excluded (abstract only) [52].

### Study characteristics

Twenty studies were prospective cohorts, and 12 were secondary analyses of RCT data (Table 1). In 31 of the 32 studies, the intervention consisted of land-based exercise therapy and education components. One study targeted weight loss in overweight and obese people with knee OA [34]. This study evaluated the association between baseline kinematics and knee pain but did not report data for age, sex, BMI, comorbidities, depression and OA severity. Seven studies were considered multidisciplinary with tailored interventions by health professionals such as dieticians or occupational therapists [10, 12, 15, 18, 30, 39, 43]. Fifteen studies were described as an OAMP, including GLA:D® [26, 41, 45, 46], BOA [24, 25, 28, 29, 44, 48], Osteoarthritis Chronic Care Program (OACCP) [10, 12, 43] and the Joint Clinic [15, 30].

Study dates overlapped for several registry-based cohort studies [24, 28, 29, 41, 44, 45, 48, 63]. Secondary analysis of RCT studies either combined participant data from multiple different exercise interventions within one study, or pooled data from several RCTs [21, 31, 35, 37, 42]. The duration of follow-up ranged from immediately post-intervention to 3-, 6- and 12-months post intervention (S1 Fig). The knee joint was reported separately in twenty-one studies, but nine studies reported the hip and knee joints together.

### Outcome measures

**Primary outcome.**   Twenty-eight studies reported various pain and function measures (Table 2). Most outcomes were self-reported, including VAS, NRS and WOMAC for pain, self-reported minimum physical activity level [29, 44, 46] and composite measures such as WOMAC-G and OKS. Three studies reported a global rating of change [10, 14, 18]. Two studies measured walking speed with the 40metre fast-paced walk test [26, 41].

**Responder definitions.**   Table 2 briefly summarises each study's "responder" definition. There was a range of different responder definitions including:

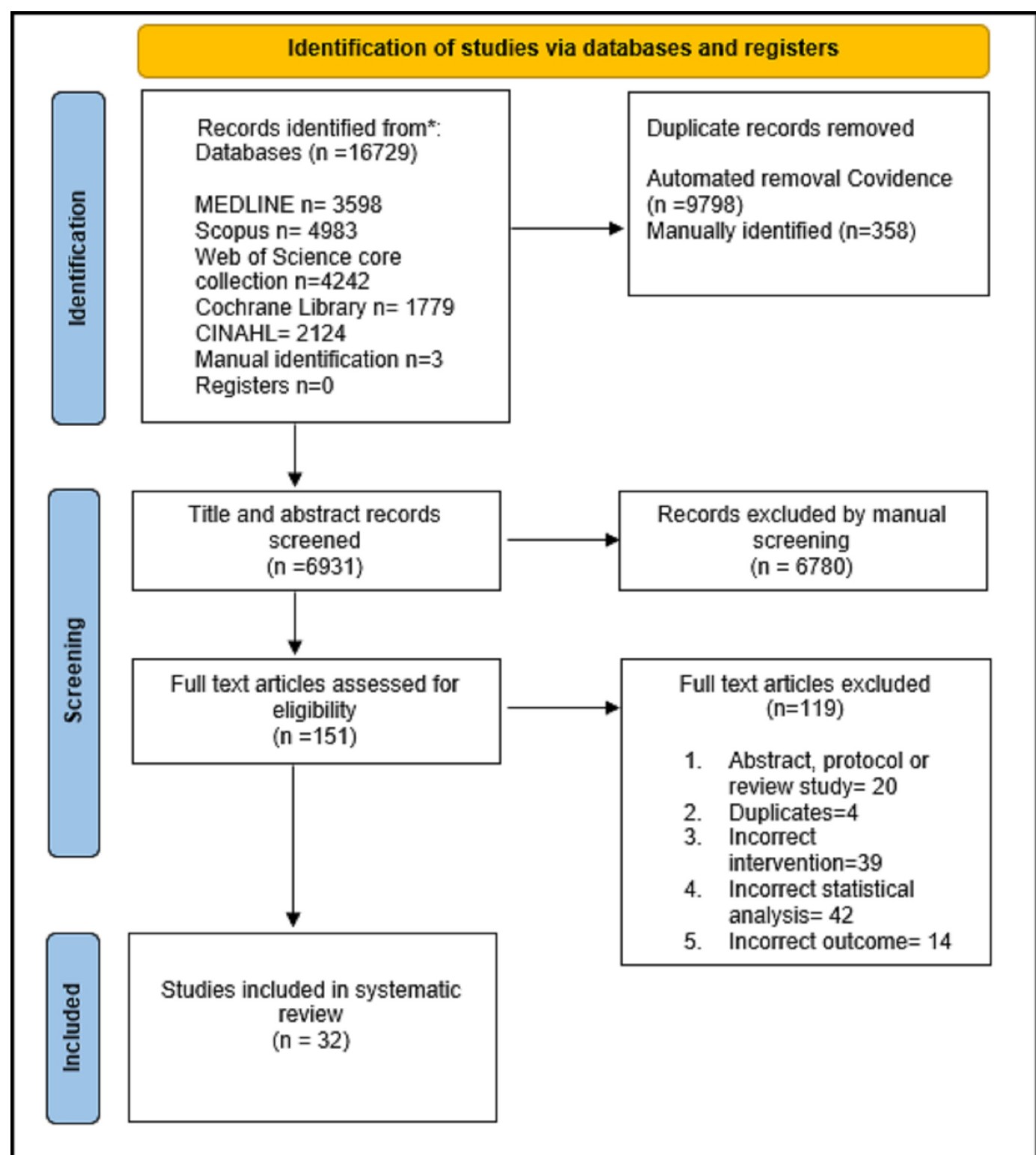

**Fig 1. PRISMA 2020 flow diagram of selected studies.**

**Table 1. Study characteristics (study design, inclusion criteria and intervention components.**

| Study | Study Design Setting and Dates | Inclusion Criteria | Joint | Participants | Intervention | Intervention Duration / Follow-up |
|---|---|---|---|---|---|---|
| Studies examining baseline participant characteristics associated with improvements in pain and function | | | | | | |
| Dell'llosa 2020 [25] | ' Register-based cohort Primary or secondary care 2008–2016 | BOA registry Clinical diagnosis hip or knee OA | Knee analysed separately | Age (knee) 66 Female 71% BMI (knee) 28.2 | 2 OA education sessions 12 group-based exercise sessions Physiotherapist supervised† | 6 weeks Follow up: 3 + 12 months |
| Degerstedt 2020 [28] | Register-based cohort Primary care 2008–2012 | BOA registry Clinical diagnosis hip or knee OA | Hip and knee OA | Age 65 Female 72% BMI 42% overweight 28% obese | 2 OA education sessions 12 group-based exercise sessions Physiotherapist supervised† | 6 weeks Follow up: 3 + 12 months |
| Ernstgard 2017 [29] | Register-based cohort Primary care 2008–2012 | BOA registry Clinical diagnosis of hip or knee OA | Hip and knee OA | Age 65 Female 75% BMI 28 | 2 OA education sessions 12 group-based exercise sessions Physiotherapist supervised† | 6 weeks Follow up: 3 + 12 months |
| Unevik 2020 [44] | Register-based cohort Primary care 2008–2017 | BOA registry Clinical diagnosis of hip or knee OA | Hip and knee OA | Age Female 71% BMI 27.8 | 2 OA education sessions 12 group-based exercise sessions Physiotherapist supervised† | 6 weeks Follow up: 3 + 12 months |
| Skou 2018 [45] | Register-based cohort Primary care 2013–2016 | GLA:D registry Clinical diagnosis of hip or knee OA | | Age 64 Female 73% BMI 28 | 2 OA education sessions 12 group-based exercise sessions Physiotherapist supervised | 6 weeks Follow up: 3 + 12 months |
| Pihl 2021 [41] | Register-based cohort Primary care 2014–2018 | GLA:D registry Clinical diagnosis of hip or knee OA | Hip and Knee OA | Age 65 Female 73% BMI 28.2 | 2 OA education sessions 12 group-based exercise sessions Physiotherapist supervised | 6 weeks Follow up: 3 + 12 months |
| Baumbach 2020 [26] | Register-based cohort Primary care 2014–2017 | GLA:D registry Clinical diagnosis of hip or knee OA | Knee OA | Age 65 Female 72% BMI 27.8 | 2 OA education sessions 12 group-based exercise sessions Physiotherapist supervised | 6 weeks Follow up: 3 months |
| Peat 2022 [46] | Register-based cohort Primary care 2014–2018 | GLA:D registry Clinical diagnosis of knee OA | Knee OA | Age 63 Female 73% BMI 28.85 | 2 OA education sessions 12 group-based exercise sessions Physiotherapist supervised | 6 weeks Follow up: 3 + 12 months |
| Eyles 2014 [12] | Prospective cohort Hospital + primary care 2011–2013 | Symptomatic or radiographical hip or knee OA diagnosis- 90% from elective TJR waitlists | Hip and knee OA | Age 69 Female 62% BMI (knee) 32.5 | Physiotherapy delivered exercise program Education about OA and comorbidities Referral to multidisciplinary services | 52 weeks Follow up: 3 + 6 months |
| Eyles 2016 [10] | Prospective cohort Hospital + primary care 2012–2014 | Symptomatic or radiographical hip or knee OA diagnosis TJR waiting list or doctor referral | Hip and knee OA | Age 66 Female 69% BMI 30 | Physiotherapy delivered exercise program Education about OA and comorbidities Referral to multidisciplinary services | 52 weeks Follow up: 6 months |
| Gwynne-Jones 2018 [15] | Prospective Cohort Hospital 2012–2014 | Patients seen at Joint Clinic not undergoing TJR or on waitlist | Hip and knee OA- | Age 69 Female 56% BMI 30.8* *Available for n = 89/218 | Multimodal exercise intervention + OA education + referral to multidisciplinary services. Six physio led group or individual sessions. | 6 months Follow up: Average 12 months |
| O'Leary 2020 [18] | Prospective cohort Hospital 2014–2016 | Triaged from orthopaedic surgical waitlist with clinical and radiological diagnosis knee OA | Knee OA | Poor response: Age 63 Female 46% BMI 34.84 Positive response: Age 62 Female 40% BMI 33.7 | Physiotherapist led multidisciplinary non-surgical intervention for knee OA Components not clearly defined included self-management, physiotherapy, and dietetics. | Unclear Follow up: 6 months. |

(*Continued*)

**Table 1.** (Continued)

| Study | Study Design Setting and Dates | Inclusion Criteria | Joint | Participants | Intervention | Intervention Duration / Follow-up |
|---|---|---|---|---|---|---|
| O'Leary 2018b [38] | Retrospective cohort Hospital 2008–2010 | Triaged from orthopaedic surgical waitlist with clinical and radiological diagnosis knee OA | Knee OA | Age 60 Female 61% BMI 33.2 | Physiotherapist led multidisciplinary non-surgical intervention for knee OA. Components not clearly defined included self-management, physiotherapy, and dietetics | Unclear Follow up: discharge from the program. |
| O'Leary 2018a [39] | Prospective cohort Outpatient physiotherapy | ACR diagnosis of knee OA Physiotherapy and orthopaedic outpatient lists | Knee OA | Age 64 Female 86% | Physiotherapy led individualised treatments (Exercise + Education about OA + weight loss advice) | 4–6 sessions Follow up: Post intervention + 6 months |
| Tanaka 2021 [17] | Longitudinal cohort Orthopaedic Rehabilitation clinic 2017–2018 | Clinical and radiological diagnosis of knee OA 50–90 years | Knee OA | Age 71 Female 82% BMI 23.9 | Physiotherapy supervised open and closed chain strengthening program 2–3 times per week + OA education | 12 weeks Follow up: 3 months |
| Weigl 2006 [14] | Prospective cohort | ACR diagnosis of OA referred by a medical practitioner | Hip and knee OA | Age 65 Female 72% BMI not measured | Individual treatment of group-based exercise therapy, patient education and coping techniques + other non-surgical, non-pharmacological interventions | 3–4 weeks Follow up: 6 months |
| Lee 2018 a [36] | Secondary analysis RCT Primary care 2010–2014 | ACR diagnosis of knee OA > 40 years | Knee OA | Age 61 Female 70% BMI 32 | Twice weekly yang style tai chi or land-based exercise program with physiotherapist Education about physical activity and exercise | 12 weeks Follow up: 3 months |
| Lee 2018 b [21] | Secondary analysis RCT Primary care 2010–2014 | ACR diagnosis of knee OA > 40 years | Knee OA | Age 61 Female 71% BMI 32 | Twice weekly yang style tai chi or land-based exercise program with physiotherapist Education about physical activity and exercise | 12 weeks Follow up: Weekly over 12 weeks |
| Lee 2017 [35] | Secondary analysis RCT Primary care 2010–2014 | ACR diagnosis of knee OA > 40 years | Knee OA | Age 60 Female 74% BMI 33 | Twice weekly yang style tai chi or land-based exercise program with physiotherapist Education about physical activity and exercise | 12 weeks Follow up: 3 months |
| Chang 2019 [64] | Secondary analysis RCT Primary care 2010–2014 | ACR diagnosis of knee OA > 40 years | Knee OA | Age 61 Female 72% BMI 32.4 | Twice weekly yang style tai chi or land-based exercise program with physiotherapist Education about physical activity and exercise | 12 weeks Follow up: 3 months |
| Legha 2020 [37] | Secondary analysis RCT Primary care 2001–2005 | Knee pain attributable to OA in primary care | Knee OA | Age 65 Female 59% BMI Overweight/ obese 79% | TOPIK (advice/exercise)[1] Apex (advice /exercise)[2] BEEP (advice/ exercise)[3] | 10 weeks[1] 10 days[2] 12 weeks[3] Follow up: 6 months |
| Quicke 2018 [42] | Secondary analysis RCT (BEEP trial) | Clinical diagnosis knee OA >45 years | Knee OA | Age 64 Female 51% BMI 39% obese 42% overweight | Physiotherapy led lower limb exercise program + education (advice and knee information booklet) Three different exercise intervention arms from original RCT with a variable number of sessions (range 4 to 10 sessions) | Variable Follow up: 3 + 6 months |
| Knoop 2014 [31] | Secondary analysis RCT Primary care 2009–2011 | ACR diagnosis knee OA Subset of original RCT | Knee OA | Age 62 Female 67% BMI 29.2 | Physical therapist progressive group exercise twice weekly + supplemented home exercise program Three education sessions on OA | 12 weeks Follow up: 3 months |

(*Continued*)

**Table 1.** (Continued)

| Study | Study Design Setting and Dates | Inclusion Criteria | Joint | Participants | Intervention | Intervention Duration / Follow-up |
|---|---|---|---|---|---|---|
| Hall 2018 [34] | Secondary analysis RCT (IDEA trial) Primary care 2006–2009 | Radiological diagnosis of knee OA >55 years BMI >27 kg/m$^2$ Those with varus thrust | Knee OA | Age 66 Female 78% BMI 33.6 | IDEA trial = combination of diet and exercise arms 3 X 60-minute exercise sessions per week supervised for the first six months Diet energy-restricted with the aim for 10% loss BW | 18 months Follow up: 18 months |
| Lawford 2021 [33] | Secondary analysis RCT (Intervention arm) Primary care 2016–2017 | NICE diagnosis of knee OA >45 years | Knee OA | Age 62.4 Female 63% BMI 31.1 | 5–10 physiotherapy delivered telephone consultations OA education and guidance of structured home strengthening program | 6 months Follow up: 6 + 12 months |
| Lawford 2018 [32] | Secondary analysis RCT (Intervention arm) Primary care | Clinical diagnosis knee OA > 50 years | Knee OA | Age 60.8 Female 56% BMI 32 | Internet-based Seven sessions 30–45-minutes duration Education material (varied) Lower limb strengthening + home exercise program (3 times per week) Physiotherapy delivered | 12 weeks Follow up: 3 + 9 months |
| Nelligan 2021 [47] | Secondary analysis RCT examining treatment moderators Primary care | Clinical diagnosis of knee OA > 45 years and had internet access | Knee OA | Intervention group Age 60 Female 58% | Self-directed internet-based exercise, OA education, and automated text message program to assist with compliance. Exercises completed at home 3 times per week. Exercises divided into 3 X 8-week blocks. | 24 weeks Follow up: 24 weeks |
| Henriksen 2022 [16] | Secondary analysis of RCT examining treatment effect modifier Outpatient clinic | Clinical diagnosis of knee OA > 50 years with BMI ≤ 35, radiologically verified | Knee OA | GLA:D intervention Age 70 Female 58% BMI 27 | 2 OA education sessions 12 group-based exercise sessions Physiotherapist supervised | 6 weeks Follow up: 9 weeks |
| Studies examining baseline patient characteristics associated with change in willingness to undertake surgery (or undertake knee joint replacement) | | | | | | |
| Teoh 2017 [43] | Prospective cohort Hospital + primary care 2012–2014 | Symptomatic or radiographical diagnosed hip or knee OA TJR waitlist or doctor referral | Hip and knee OA | Age 66 Female 69% BMI ≤ 30 (4%) | Physiotherapy delivered exercise program Education about OA and comorbidities Referral to multidisciplinary services | 52 weeks Follow up: 12 months or final review |
| Dell'lsola 2021 [24] | Register-based cohort Primary care 2008–2016 | BOA registry Clinical diagnosis of hip or knee OA | Hip and knee OA Knee OA analysed separately | Age 66 Female 69% BMI 28 | 2 OA education sessions 12 group-based exercise sessions Physiotherapist supervised† | 6 weeks Follow up: 3 + 12 months |
| Gustafsson 2022 [48] | Register-based cohort Primary care 2008–2016 | BOA registry Clinical diagnosis of hip or knee OA | Hip and knee OA Knee OA analysed separately | Age 66 Female 69% BMI 28 | 2 OA education sessions 12 group-based exercise sessions Physiotherapist supervised† | 6 weeks Follow up: 3 + 12 months |
| Gwynne-Jones 2020 [30] | Prospective cohort Hospital 2012–2014 | Patients seen at Joint Clinic not undergoing TJR or on waitlist | Hip and knee OA Knee OA analysed separately | Age 68 Female 55% BMI 31.5 | Multimodal exercise intervention + OA education + referral to multidisciplinary services. Six physio led group or individual sessions. | 6 months Follow up: 5 years |

† The BOA intervention is described as a supported self-management programme. Not all participants take part in a supervised rehabilitation setting. The 2013 registry figures report that 80% of participants opt for an individual programme, and 60% participate in a supervised rehabilitation program. The BOA-Register. Better Management of patients with osteoarthritis. Annual report 2013 www.boaregistret.se.

**Table 2. Study outcomes and statistical methods (including prognostic factors, sample size, modelling method and sets of adjustment factors).**

| Study | Outcome Measure | Identification of Candidate Predictors | Initial Candidate Predictors (Number and Type) | Sample Size Missing Data | Modelling Method | Selection Criteria (multivariable modelling) | Effect Estimate | Adjustment Factors (covariates) |
|---|---|---|---|---|---|---|---|---|
| Studies examining baseline participant characteristics associated with improvements in pain and function | | | | | | | | |
| Dell'Isola 2020 [25] | Change in pain NRS (0–10) mean pain intensity last week | Exploratory | n = 20 Baseline pain, demographics, Self-efficacy (ASES) Willingness for surgery (Y/N) Drug intake Charnley classification Fear of movement (Y/N) Physical activity, treatment modality, QOL, previous surgery | 23 309 (analysed) 20 919 (missing) 44 228 (eligible) | Multivariable linear regression | Full model fitted + stepwise selection methods Variables with p ≥ 0.2 excluded, change in > 10% retained as confounder + variables p > 0.05 and < 0.2 | Unstandardised regression coefficient (ß) (Adjusted) | Retained as a confounder if exclusion changed the estimate by > 10% |
| Degerstedt 2020 [28] | Change in pain and physical activity (PA) VAS (0–100) average pain last month Self-reported PA (days per week > 30 minutes) | A priori Main prognostic factor of interest = baseline self-efficacy | n = 10 Baseline self-efficacy (ASES) Baseline pain, physical activity, demographics, BMI, affected joint, walking difficulty | 3266 (analysed) 352 (missing) | Univariable + mixed model linear regression Random effects for time | Full model fitted 10 confounders added based on univariable screen (p < 0.05) | Least square mean difference (Adjusted) | Age, sex, birthplace, education, marital status, affected other joints, most affected joint, BMI, walking difficulty + duration of intervention |
| Ernstgard 2017 [29] | Change in PA PA self-reported Responder = > 30 minutes daily or 150 minutes per week | A priori based on expert opinion and evidence Main prognostic factors of interest = age, BMI, sex, and comorbidities. | n = 5 Age, sex, Charnley Category BMI, time | 6810 (analysed) 3845 (missing) 10455 (eligible) | Mixed model logistic regression (GEE model) | Full model fitted with age, sex, BMI, comorbidities, and time | OR (Adjusted) | Age, sex, BMI, comorbidity, time |
| Unevik 2020 [44] | Change in pain, function, and willingness to undertake surgery NRS pain (0–10) PA > 150 minutes (Yes/No) Difficulties walking (Yes/No) Willingness to undertake surgery (Yes/No) | A priori Main prognostic factor of interest = Sociodemographic determinants | n = 5 Level of education (compulsory, secondary, university) Born in Sweden (Yes/No) | 22 741 (analysed) 72 057 (missing) 94798 (eligible) | Multivariable logistic regression | Full model with PF of interest (sociodemographic) + prespecified confounders | OR (Adjusted) | Age, sex, BMI + baseline values |
| Skou 2018 [45] | Change in pain VAS pain (0–100) last month | A priori Main prognostic factor of interest = physical activity level | n = 2 Physical activity (UCLA) Physical activity level (self-reported question) | 12 796 (analysed) Final analysis included those with baseline + ≥ 1 follow up measure | Mixed model linear regression Random effects for clinics, patients, and time | Univariable and multivariable models Full model was fitted separately for PF of interest + prespecified confounders | Mean difference (Adjusted) | Prespecified Age, sex, BMI, educational level, and comorbidity index |
| Pihl 2021 [41] | Change in pain and function 40m Fast-paced walk test VAS (0–100) | A priori Main prognostic factor of interest presence of comorbidities | n = 33 20 most common comorbidities | 24 513 (analysed) 8577 (missing) | Mixed Linear regression Random effects clinics + groups | All PF of interest in full model (comorbidities) and adjusted for prespecified confounders (13 preidentified confounders from DAG (Directs Acyclic Graph) | Mean difference (Adjusted) | Prespecified Age, sex, BMI, educational level, other demographics, other joint pain, self-reported PA, use of analgesics |

*(Continued)*

**Table 2.** (Continued)

| Study | Outcome Measure | Identification of Candidate Predictors | Initial Candidate Predictors (Number and Type) | Sample Size Missing Data | Modelling Method | Selection Criteria (multivariable modelling) | Effect Estimate | Adjustment Factors (covariates) |
|---|---|---|---|---|---|---|---|---|
| Baumbach 2020 [26] | Change in pain and function 40m Fast-paced walk test VAS (0–100) | 222 potential variables at baseline (excluded if > 70% missing data, irrelevant predictors) | 51 patient characteristics were selected based on best performing variables | 6767 (analysed) 8075 (missing) 14 824 (eligible) | Random forest regression Linear regression | Variable importance based on RMSE (root mean squared error) Variable received score of 0–100 (least to most important) Elbow method | Performance of model assessed using $R^2$ + RSME | N/A |
| Peat 2022 [46] | Change in pain VAS (0–100) average pain and maximum pain last month | A priori Main prognostic factor of interest social disadvantage | n = 1 Social disadvantage dichotomised based on 3 social stratifies (education, place of birth and citizenship) | 12 493 (classified higher social advantage) 250 (classified lower social advantage) 24–25% outcomes missing 3 months 38–39% missing outcomes 12 months 18 848 (eligible) | Univariable and multivariable linear regression | Crude differences were reported between groups and subsequent adjustments for groupings that had been preidentified | MD (Adjusted) | Prespecified 7 different groupings Baseline outcomes, treatment centres, demographics, comorbidities, BMI, psychological factors, previous treatment and attendance |
| Eyles 2014 [12] | Change in WOMAC-G Responder = MCID 18% change or 9 points from baseline | A priori Based on literature review and expert opinion. Identified 8 PF of interest | n = 8 Age, sex, BMI, baseline pain VAS, depression (DASS21), signal joint, 6MWT, comorbidities (SCQ) | 308 (analysed) 251 (missing) 559 (eligible) | Univariable + multivariable logistic regression | Full model fitted with all predictor variables Backward elimination technique (least significant predictor removed) | OR (Adjusted) | Covariate retained if coefficient changed by > 10% All variables except signal joint and sex were retained |
| Eyles 2016 [10] | Three definitions of non-responder 1. WOMAC-G (MID 9.6 points or 21%) 2. Transition scale (moderately or much worse) 3. Combination of WOMAC-G and transition scale | A priori Based on literature review and expert opinion. Identified 9 PF of interest | n = 9 Age, sex, BMI, baseline pain (VAS), signal joint, depression (DASS21), Comorbidities (SCQ), 6MWT, TJR waitlist | 386 (analysed) 153 (missing) 539(eligible) | Univariable + multivariable logistic regression | Variables with p< 0.2 included in multivariable model. Backward elimination (least significant predictor removed) | OR (Adjusted) | Covariate retained if coefficient changed by > 10% upon removal from model |
| Gwynne-Jones 2018 [15] | Change in pain and function Oxford knee scale (OKS) + SF-12 Responder = improvement > MCID for each score (MCID > 5 points) | Exploratory Factors at baseline as part of program | n = 5 Age, Sex, Affected joint BMI, Oxford knee score SF-12 PCS SF-12 MCS | 218 (analysed) 26 (missing SF-12 scores only) | Multinomial linear + logistical regression | Full model fitted with affected joint + baseline PROM Adjusted for prespecified confounders | OR Mean difference (Adjusted) | Prespecified confounders–age, gender, and BMI Rerun with adjustment for time to follow up |

(*Continued*)

**Table 2.** (Continued)

| Study | Outcome Measure | Identification of Candidate Predictors | Initial Candidate Predictors (Number and Type) | Sample Size Missing Data | Modelling Method | Selection Criteria (multivariable modelling) | Effect Estimate | Adjustment Factors (covariates) |
|---|---|---|---|---|---|---|---|---|
| O'Leary 2020 [18] | Global rating of change (GRoC) Responder (+2 to +7) & non responder (-7 to +1) on 15-point Likert scale | Exploratory Baseline data collected at time of initial consultation | n = 20 Demographics, general health, psychological measures, clinical factors Condition-specific symptoms and signs (example, KOOS, radiological severity) | 238 (analysed) 48 (missing) 286 (eligible + consented) | Univariable + mixed model multivariable logistical regression Random effects for clinical site | Full model fitted from univariable screen (p ≤ 0.1) + Multicollinearity screen (r >0.4 screened) + clinical judgement | OR (Adjusted) | No prespecified confounders Full model fitted (univariable + multicollinearity screen) |
| O'Leary 2018b [38] | Change in pain and function (WOMAC) Responder/ non-responder = MCID ≥ 10-point change in WOMAC | Exploratory Baseline data collected at time of initial consultation | n = 27 Demographics, QOL (AQoL-6D), psychological measures (DASS21, Pain self-efficacy ASES) VAS, patient specific functional scale, comorbidity (number), Radiological findings (absent/mild/ moderate/ severe) | 190 (knee OA analysed) Missing data not reported 631(eligible) | Univariable + multivariable logistical regression | Full model fitted from univariable screen (p of ≤ 0.1 included) + backward elimination technique | OR (Adjusted) | No preidentified confounders Full model fitted |
| O'Leary 2018a [39] | Change in pain and function (WOMAC) OMERACT-OARSI responder definition Global rating of change (GRoC) | A priori Based on literature and clinical judgement Main PF of interest = pain sensitisation | n = 20 Qualitative sensory testing (12 variables) Demographics, comorbidity score (SCQ), depressive symptoms (CES-D), pain (NRS) Central sensitisation inventory | 99 (analysed 6/12) 57 (missing) 156 (enrolled) 387 (eligible) | Univariable + multivariable model (hierarchical model) | Full model fitted based on univariable screen (P ≤ 0.1) + preidentified predictors | OR (Adjusted) | Preidentified based on literature—age, sex, depression, treatment adherence and comorbidities |
| Tanaka 2021 [17] | Change in pain and function NRS 0–10 and OKS Responder = reduced pain by > 50%, and OKS reduced ≥ 5 points | Exploratory Hypothesis generating, based on previous literature | n = 11 Age, sex, BMI Pain duration, Medication use, KL grade, Pain catastrophising (PCS) Pain self-efficacy (PSEQ 0–60) Knee Body function (FreKAQ) | 150 (analysed) 127 (missing) 277 (eligible) | Multivariable Logistic regression + Classification and Regression Decision Tree | Full model with all 11 predictor variables | OR (Adjusted) | Not stated |

(*Continued*)

**Table 2.** (Continued)

| Study | Outcome Measure | Identification of Candidate Predictors | Initial Candidate Predictors (Number and Type) | Sample Size Missing Data | Modelling Method | Selection Criteria (multivariable modelling) | Effect Estimate | Adjustment Factors (covariates) |
|---|---|---|---|---|---|---|---|---|
| Weigl 2006 [14] | Three definitions responder 1. WOMAC-G MCID >18% 2. Transition scale (improvement in health 3. Combination of WOMAC-G and transition scale | Exploratory All baseline predictors initially examined | n = 21 Demographics, Comorbidities (SCQ) Lifestyle risk factors Depression + Anxiety (HADS (Hospital Anxiety Depression Score)), SF36 Sense of coherence (SOC) | 250 (analysed) 14 (missing) 264 (eligible) | Sequential logistic regression | 4 step process Univariable screen (P < 0.2) Forward and backward stepwise selection Remained in model if AUC (Area Under Curve) increased 5% or significant predictor | OR (Adjusted) | No preidentified confounders Predictors identified by a 4-step data driven process |
| Lee 2018 a [36] | Change in pain and function (WOMAC) Responder = ≥ 50% improvement WOMAC | A priori Based on research and discussion | n = 20 Demographics (age, sex, race) BMI, WOMAC pain + function Duration of pain, patient global assessment, SF-36, CHAMPS PA scale, Sleep disturbance, Depression (BeckII), Perceived Stress Scale, five facet mindfulness scale Self-efficacy (ASES) Comorbidities (self-report) Outcome expectations | 182 (analysed) 22 (missing) 204 (parent RCT) | COX proportional hazards multivariable regression | Full model fitted based on univariable screen (P < 0.05 from unadjusted models) | HR Only unadjusted reported | Not clear |
| Lee 2018 b [21] | Pain and function trajectories WOMAC (measured weekly over 12 weeks) | Exploratory Baseline variables measured in previous RCT identified a priori with consideration to plausibility from OA literature | n = 30 Demographics, WOMAC pain function, Depression (BeckII) Self-Efficacy (ASES), Other PROMs (physical & psychosocial health), Physical performance tests (6MWT, leg extensor strength, Berg balance) Comorbidities (self-report) KL grade/ radiographical alignment | 171 (analysed) 33 (excluded) | Multinomial logistic regression Group-based modelling identified four different pain and function trajectories | Preselected PF added to full model and adjusted for age, sex, and BMI Separate model for all four identified pain trajectories (univariate analysis) | OR (Adjusted) | Preidentified confounders of age, sex, BMI |

(Continued)

**Table 2.** (Continued)

| Study | Outcome Measure | Identification of Candidate Predictors | Initial Candidate Predictors (Number and Type) | Sample Size Missing Data | Modelling Method | Selection Criteria (multivariable modelling) | Effect Estimate | Adjustment Factors (covariates) |
|---|---|---|---|---|---|---|---|---|
| Lee 2017 [35] | Change in pain and function (WOMAC) Responder criteria using OMERACT-OARSI | A priori Main prognostic factor of interest = mindfulness | Five Facet Mindfulness Questionnaire (FFMQ) Demographic and clinical characteristics | 86 (analysed) 118 (missing PF from parent RCT) | Univariable | N/A Univariable model only | RR (Unadjusted) | Attempted to adjust for confounders but reported non-significant. Did not fit a multivariable model |
| Chang 2019 [64] | Change in pain and function (WOMAC) Responder criteria using OMERACT-OARSI | A priori Main prognostic factor of interest = preintervention physical activity | Age BMI Sex PA (CHAMPS) | 166 (analysed) 48 (missing data RCT) 204 (parent RCT) | Multivariable logistic regression | Full model fitted with PF of interest + adjusted for prespecified confounders | OR (Adjusted) | Prespecified confounders- Age, sex, and BMI |
| Legha 2020 [37] | Change in pain and function (WOMAC pain) (WOMAC function) | A priori Main prognostic factor of interest = Prescence of comorbidities | n = 7 BMI, pain > 1 body site Anxiety/depression (EQ-5D) Prescence of comorbidities (Yes/no) No. comorbidities (0, 1–2, 3 +) | 1083 (from 3 parent RCT) <20% missing data at 6/12 and < 3.3% missing covariate data | Mixed linear model Individual Participant Data 3 RCT interventions | A priori selection of 6 comorbidities +number of comorbidities (0, 1–2, 3+) Each PF of interest tested separately with treatment effect + adjusted for other covariates | Mean differences (Adjusted) | Prespecified confounders- Baseline WOMAC pain or function, age, gender, intervention allocation |
| Quicke 2018 [42] | Change in pain and function (WOMAC pain) (WOMAC function) Responder criteria using OMERACT-OARSI | A priori Based on literature and expert opinion Main prognostic factor of interest = change in PA | n = 8 Change in PA, age, sex, BMI, Baseline pain and function (WOMAC) Depression (PHQ8) | 514 (analysed from parent RCT) 12–22% missing outcome data and 17% lost to follow up at 6/12 | Linear and logistic regression | Full model fitted with preidentified confounders + PF of interest. Backward elimination (p>0.05 excluded). Retained PA and baseline OA severity in full model. | OR (Adjusted) | Prespecified confounders- age, sex, BMI, demographics, anxiety/depression, widespread pain + adjustment for intervention arm |
| Knoop 2014 [31] | Change in pain and function WOMAC physical function NRS (0–10) Responder criteria using OMERACT-OARSI | Exploratory A prior Main prognostic factor of interest = baseline MRI features | n = 14 Four baseline MRI features, cartilage integrity, BMLs (Bone Marrow Lesions), osteophytes and meniscal tears in TF and PF joint | 95 (analysed) 21(missing/excluded) 112 (parent RCT) | Linear and logistical regression | Full model fitted with PF of interest (baseline MRI features) + potential preidentified confounders | OR Regression coefficients (Adjusted) | Age, sex, duration symptoms, BMI, anxiety/depression, muscle strength, malalignment, joint laxity, pain medication, proprioception accuracy |
| Hall 2018 [34] | Change in pain (WOMAC pain subscale) | A priori Based on literature/ expert opinion Main prognostic factor of interest = baseline biomechanical factors | n = 3 Frontal plane knee kinematics Peak varus velocity, frontal plane excursion + frontal plane angle | 387 (analysed with varus thrust) 454 (parent RCT) | Univariable + multivariable linear regression | Full model fitted with PF of interest (baseline biomechanical factors) + preidentified confounders | Unstandardised regression coefficient (Adjusted) | Preidentified confounders Sex, baseline BMI, age, pain, walking speed, knee alignment category, KL grade + treatment category |

(*Continued*)

**Table 2.** (Continued)

| Study | Outcome Measure | Identification of Candidate Predictors | Initial Candidate Predictors (Number and Type) | Sample Size Missing Data | Modelling Method | Selection Criteria (multivariable modelling) | Effect Estimate | Adjustment Factors (covariates) |
|---|---|---|---|---|---|---|---|---|
| Lawford 2021 [33] | Change in pain and function NRS 0–10 pain last week WOMAC physical function Global rating of change (GRoC for pain and function) | A priori- literature and expert opinion Main prognostic factor of interest = therapeutic alliance | n = 10 Therapeutic alliance (WAI (Working Alliance Inventory)) rated by patient and therapist Treatment expectations Physical therapist characteristics Self-efficacy | 87 (analysed from intervention arm RCT) 5 (missing) | Mixed model multivariable linear + logistic regression | Full model with PF of interest (therapeutic alliance) and adjusted for other PF + random effects for timepoints | Unstandardised regression coefficient + OR (Adjusted) | Baseline outcome measures gender, age, self-efficacy, treatment expectations and physical therapist characteristics (level of experience) |
| Lawford 2018 [32] | Change in pain and function NRS pain 0–10 (for walking) WOMAC physical function | Previous research and theoretical plausibility | n = 8 Age, sex, BMI, education level, employment, pain self-efficacy (ASES), pain catastrophising (PCS), expectation of treatment benefit (5-point Likert) | 148 (analysed from parent RCT) | Linear regression model | Full model with PF of interest and study group as covariates and interaction term between the two | OR + regression coefficient examining the effect of treatment (intervention-control difference) | Full model fitted with all variables of interest |
| Nelligan 2021 [47] | Change in pain and function NRS 0–10 pain last week WOMAC physical function | Exploratory Moderators were selected a priori based on literature, expert opinion and consensus | n = 5 treatment moderators Comorbidities (number) No of joints Pain self-efficacy (ASES) Exercise self-efficacy Self-perceived importance of exercise | 206 (parent RCT) >85% had pain and function outcome at follow up in control and intervention group | Linear regression | Full model fitted with 5 potential moderators | MD (Adjusted) | Full model fitted with all moderators of interest |
| Henriksen 2022 [16] | Change in pain KOOS pain subscale 0–100 | Exploratory Moderators identified a priori | n = 11 treatment moderators BMI (dichotomised) Swollen knee Radiographical severity (KL grade) Sports participation Demographics Treatment preference Analgesic use Pain (ICOAP questionnaire) Treatment preference | 206 (parent RCT) 102 (GLAD group) 13 (missing/ lost to follow up GLAD group) | ANOVA Repeated mixed linear models adjusted for stratification factors | ANOVA used with group and the moderators with their interaction and baseline KOOS pain levels as covariate Planned analysis to look at association between baseline factors and outcome (not conducted) | Mean was reported for GLAD intervention and subgroups | N/A |

Studies examining baseline patient characteristics associated with change in willingness to undertake surgery (or undertake knee joint replacement)

*(Continued)*

**Table 2.** (Continued)

| Study | Outcome Measure | Identification of Candidate Predictors | Initial Candidate Predictors (Number and Type) | Sample Size Missing Data | Modelling Method | Selection Criteria (multivariable modelling) | Effect Estimate | Adjustment Factors (covariates) |
|---|---|---|---|---|---|---|---|---|
| Teoh 2017 [43] | Change in willingness to undertake surgery 5-point Likert scale (willing to unwilling) Dichotomised outcome willing/not willing | Exploratory A priori Based on literature and expert opinion | n = 17 Demographics, age, sex, BMI, signal joint KOOS/HOOS Comorbidities (number) Depression (DASS21) 6MWT | 409 (analysed) 232 (missing) 641 (eligible) | Univariable + multivariable logistic regression | Univariable screen p < 0.25 included in final multivariable model Backward elimination (least significant predictor removed) | OR (Adjusted) | Covariate retained if coefficient changed by > 20% |
| Dell'Isola 2021 [24] | Change in willingness to undertake surgery (dichotomised yes/no question) | Exploratory Baseline data collected as part of registry | n = 17 Age, sex, BMI, demographics, comorbidities (number), physical activity, Charnley classification, fear of movement, walking difficulties, surgery, medication, previous radiographs | 30 578 (recorded baseline willingness to undergo surgery hip + knee) 20 649 (analysed knee) 20 919 (excluded + lost to follow up) 51,627 (eligible) | Multivariable logistic regression Separate models for 2 different timepoints (3 + 12 months) | Full model with PF of interest Model 1 at 3 months looked at association between pain, walking difficulties, self-efficacy, fear of movement and willingness to undergo surgery Model 2 as above at 12 months | OR (Adjusted) | Age, sex BMI, willingness to undertake surgery at previous visit, pain at previous visit, previous surgery and education |
| Gwynne-Jones 2020 [30] | Undertook TJR surgery (Yes/No) | Exploratory Baseline data collected at time of initial consultation | n = 8 Age, sex, BMI, affected Joint KL grade Oxford hip/knee (OHKS) SF-12 | 339 (analysed) 29 (missing) | Univariable and multivariable logistic regression | All baseline predictors in final model | HR (Adjusted) | All variables in the final multivariable model BMI dropped due to missing data |
| Gustafsson 2022 [48] | Time to arthroplasty at one and five years (knee joint analysed separately) | Exploratory Baseline data collected as part of registry | n = 18 Age, sex, BMI, demographics, comorbidities (number), physical activity, QOL, Charnley classification, fear of movement, pain frequency and intensity, walking difficulties, surgery, medication | 49 366 (analysed) 6 387 (missing) | COX proportional hazards multivariable regression | All baseline predictors in final model | HR (Adjusted) | All variables in the final multivariable model |

- The OMERACT-OARSI responder criteria- a composite measure that uses improvements in pain, function and the patient's global assessment of improvement [65].

- The patient's global assessment of improvement in pain or function (Likert scale). The scale is dichotomised to reflect a responder or non-responder cut-off value [18].

- Patient-reported outcome measures such as WOMAC pain or function or WOMAC-global. A range of cut-off points were based on a minimally clinically important difference (MCID).

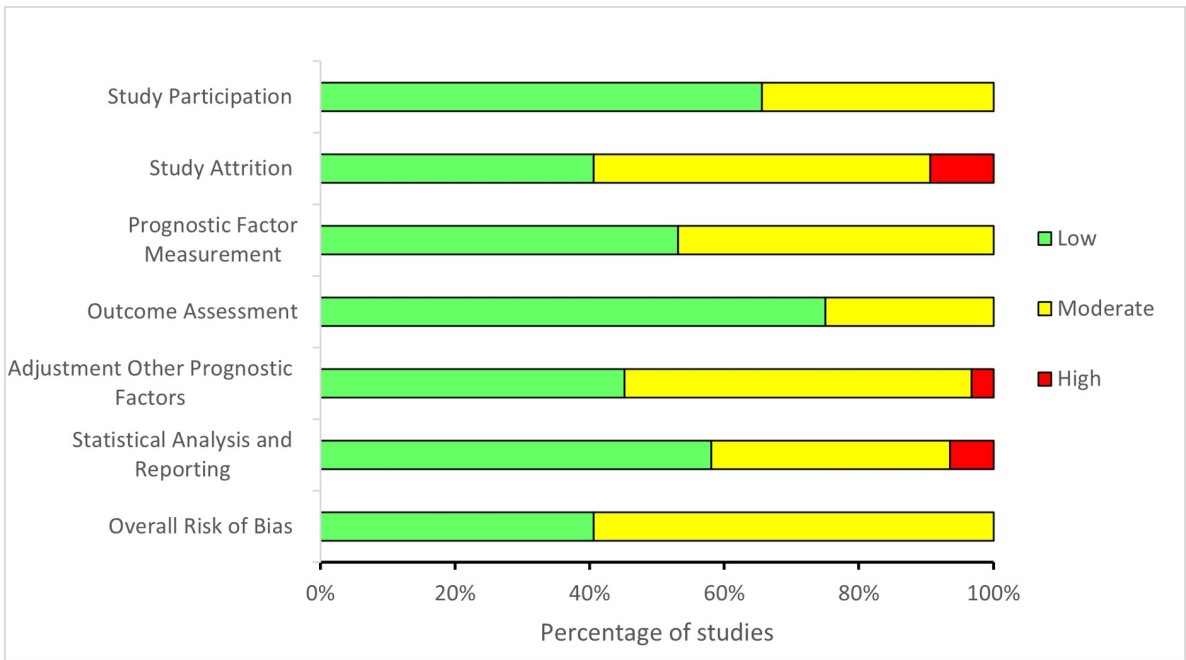

**Fig 2. Summary of risk of bias assessment using Quality in Prognostic Studies (QUIPS) according to the six domains of the 32 included studies.**

- Multiple responder definitions constructed within a single study which produced several similar effect measures [10, 12, 14, 39].

- Pain scales such as NRS pain (0–10) were dichotomised. A responder was defined as having greater than 50% reduction in pain following the intervention [17].

## Secondary outcome

Four studies reported change in willingness to undertake joint surgery or undertake total knee joint replacement (secondary objective) [24, 30, 43, 48]. Two studies reported hazard ratios and baseline characteristics associated with time to joint replacement surgery [30, 48] and two studies reported change in willingness to undergo surgery [24, 43]. Change in willingness to undertake surgery was assessed differently by the studies. One study asked the question "Are your knee/hip symptoms so severe that you wish to undergo surgery? (Yes/No) [24] and one study used a 5-point scale rating willingness for surgery (from 'definitely willing', 'probably willing', 'unsure', 'probably unwilling' and 'definitely unwilling') [43] (S5 Table).

## Prognostic factor identification

There was considerable variation in the studies' methods of identifying prognostic factors. Many studies are considered exploratory as they evaluated many baseline prognostic factors collected from a registry, multidisciplinary program, or a previous RCT study [14, 15, 18, 21, 37, 38, 44]. Several studies identified prognostic factors a priori based on expert opinion and literature review [10, 12, 28, 29, 31, 33–35, 37, 39, 42, 44].

Adjustment for other prognostic factors varied between studies. Several studies implemented prespecified confounders into their model design. The most common adjustment

factors were age, sex, and BMI [15, 21, 28, 29, 31, 33, 34, 37, 39, 41, 42, 44, 45, 64]. A univariable screen was used in 9 studies [10, 14, 18, 21, 28, 36, 38, 39, 43] (Table 2).

## Risk of bias

Most studies were rated as moderate overall risk of bias (20/32), and the remainder (12/32) had a low overall risk of bias. Almost half of the studies did not adequately account for adjustment for other prognostic factors, with 16 rated as moderate and one as high risk of bias. Most studies had a low risk of bias for statistical analysis and reporting (18/32), while the remainder (14/32) were rated as moderate or high. Study attrition was rated as moderate or high in 19 of the 32 studies; however, there was difficulty interpreting the loss-to-follow-up and response rate in large registry-based cohort studies [58]. A summary of the risk of bias assessment is shown in Fig 2 and ratings for the individual studies in S4 Table.

## Prognostic factor results

Studies reported a wide range of prognostic factors. Table 2 includes the number and types of prognostic factors that were evaluated.

The exact number of potential prognostic factors of interest was hard to determine in exploratory studies that may report up to 20 potential prognostic factors [14, 18, 21, 25, 38]. Studies that reported a single prognostic factor of interest included self-efficacy [28], comorbidities [37, 41], demographics [44], pain sensitisation [39], mindfulness [35], physical activity [42, 45], baseline imaging [31], baseline biomechanical factors [34] and therapeutic alliance [33].

Due to the wide variation in reported factors, this review has focussed on the pre-identified factors of age, sex, BMI, depression, comorbidity and OA severity with a descriptive summary and visualisation of the odds of a positive response (OR and 95% CI). These factors have been examined in studies examining predictors of total knee joint replacement and OA progression but less commonly in predicting a response to first-line interventions [66, 67]. Effect measure results (OR, MD, HR, Beta coefficient (ß)) and rescaling calculations for individual studies and prognostic factors are shown in S5 Table.

**Studies examining baseline OA severity (imaging).** Baseline OA severity (imaging) was reported in four small exploratory studies [16, 18, 31, 68]. Comparison between studies was difficult because each used different imaging modalities, outcome measures and cut-off points for grading OA severity (S5 Table).

Knoop (2014) found all grades of OA severity on MRI were associated with a positive response following an exercise and education intervention, but found the response was reduced with advance PF OA [31]. O'Leary (2020) found severe medial compartment OA was associated with a poorer response compared to mild medial compartment OA. OA severity was assessed using radiological, CT or MRI results reported in the medical records. Severity was recorded in the medial, lateral and PF compartments as either absent, mild, moderate or severe. Lee (2018) examined the association between radiological KL grade and four different pain and function trajectory groups over 12 weeks. It was found KL grades were evenly distributed amongst the four groups and the authors reported no significant association between KL grade and trajectory group membership.

**Effect of increasing age on the odds of positive response to intervention.** We found moderate certainty evidence that older age may be associated with a lower odds of responding to a combined first-line intervention (Fig 3). The effect estimates were small, precise, and slightly negative (less than 10%) in studies that reported age continuously (OR = ~0.9) (Fig 3). One study was imprecise but still reported a negative association between increasing age and the probability of a positive response (unadjusted OR 0.9, 95% CI 0.7–1.2) [12].

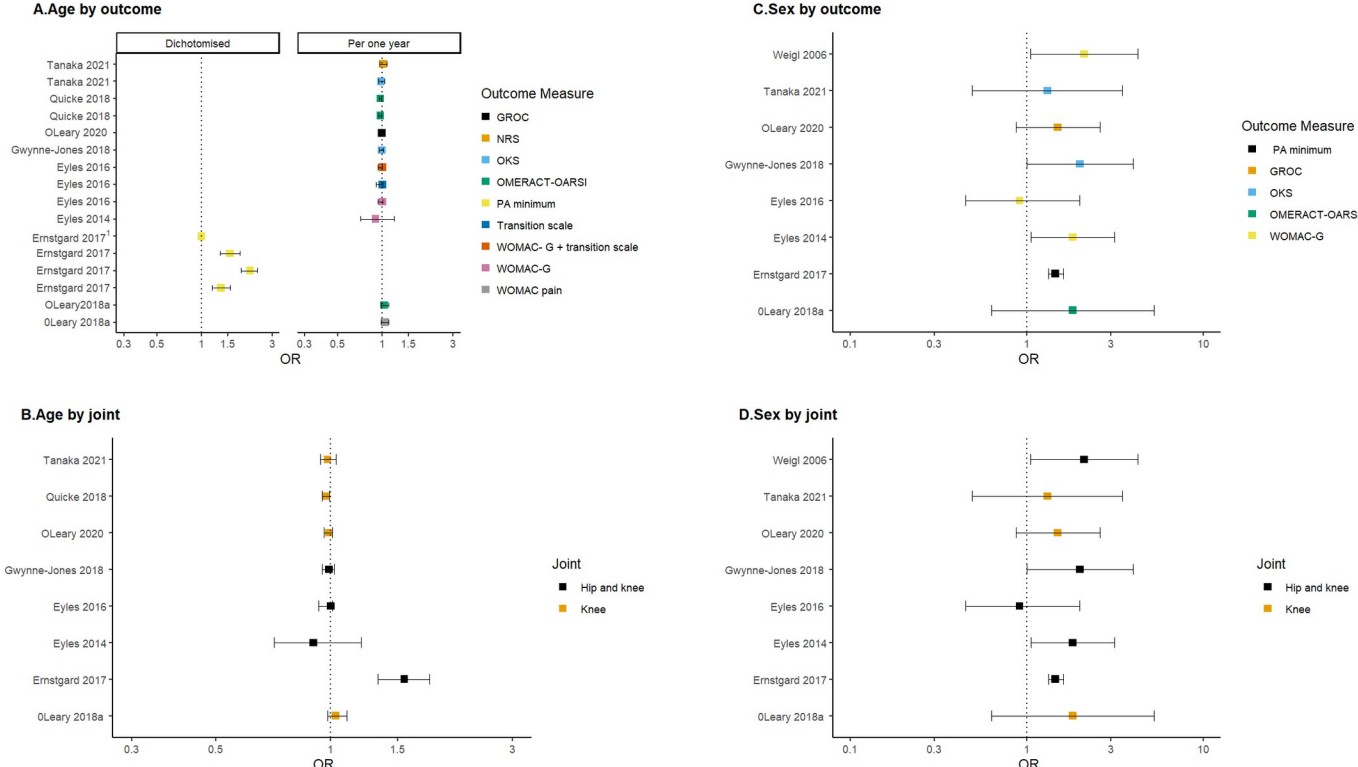

**Fig 3. Effect of increasing age and female sex on the odds of positive response following a combined first-line intervention for knee osteoarthritis.** All graphs report the log odds ratio and 95% CI. Repeated study labels by the same author represent multiple responder definitions within each study. The OR for sex reports the probability of a female (compared to a male) being a responder. OR > 1 = increased probability of female being a responder compared to a male. OR > 1 for age interpreted as increased probability of being a responder with increasing age. For age, original data from 8 studies reporting OR (7 cohorts, one secondary analysis of RCT). Studies not included: 2 reporting regression coefficients [25, 42] and 3 MD [15, 16, 37] and 1 HR [36]. For sex, original data from 8 cohort studies reporting OR. Studies not included: 3 reporting MD [15, 16, 37] and 1 HR [36]. 1. Ernstgard 2017 is grouped by ages and represents the OR of 75–100 years, 65–74 years, 55–64 years compared to 22–54 years [29].

The results for Ernstgard (2017) differed. This study examined a physical activity measure, with a responder defined as exceeding a self-reported minimum physical activity threshold of 150 minutes per week or greater than 30 minutes on four or more days per week [29]. Age was not reported continuously with four aged groups compared (22–54, 55–64, 65–74 and 75 + years) [29]. Ernstgard (2017) found being older was associated with up to twice the odds of positive response [25]. The OR for 65–74 years compared to those aged 22–54 years was 2.13 (95% CI 1.85–2.38), suggesting that older people were more likely to be physically active than younger people. Comparing the results of this study is difficult due to the different outcome measures and that age was not reported continuously. Additionally, only a small proportion of patients reported a change in the minimum physical activity threshold (a slight increase from 77% at baseline to 82% at three months and decreased to 76% at 12 months) [29].

**Effect of female sex on the odds of positive response to intervention.** Low certainty evidence indicated that being female was associated with a positive response following a combined first-line intervention (Fig 3). Females had up to 2–3 times the odds of a positive response compared to males. The effect estimates for females (compared to males) were positive but imprecise in 7 out of 8 studies (OR ranging between 1 and 3). Four of these studies included small cohorts of less than 300 participants [14, 15, 17, 18, 39]. Weigl (2006) reported the three largest effect estimates from three different responder definitions. The OR using WOMAC-G responder (based on MCID 18% improvement) was 2.11 (95% CI 1.05–4.25)

**A. Depression by measure and cut off point**

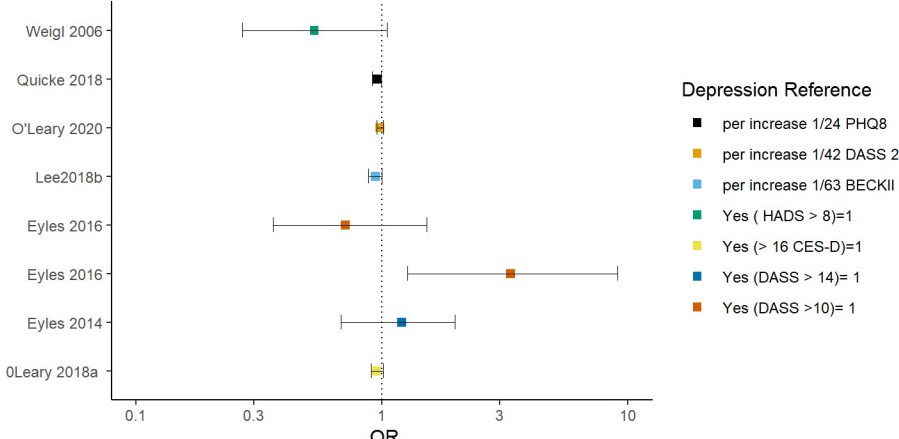

**B  BMI by model type and cut off point**

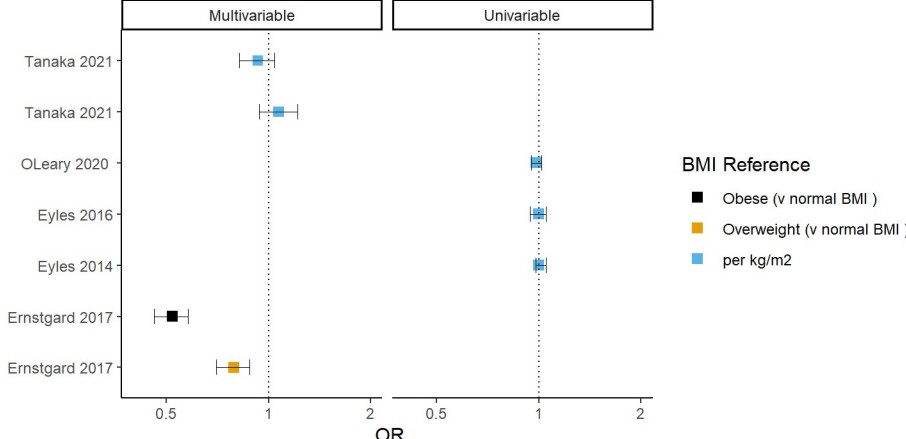

**Fig 4.  Effect of depression and increased BMI on the odds of positive response following a combined first-line intervention for knee osteoarthritis.** Graph reports the log odds ratio and 95% CI. Repeated study labels by the same author represent multiple responder definitions within each study. Graph plots the odds ratio for the probability of BMI or presence of depression on a positive response following intervention. For BMI, OR >1 = increased probability of being a responder with increased BMI. For depression, OR > 1 = increased probability of being a responder with the presence of depression. For a continuous predictor, we interpret the odds ratio per one unit change, and for a dichotomised predictor, the OR is the probability compared to the reference group. For depression, original data from 7 studies (5 cohorts) reporting OR. Studies not included: 1 reporting a regression coefficient [42] and 2 mean difference [37, 41]. For BMI, original data from 5 cohort studies reporting OR. Studies not included: 1 reporting regression coefficient [25], 2 MD [16, 37] and 1 unadjusted HR [36]. HADS = Hospital Anxiety Depression scale, PHQ-8 = Patient Health Questionnaire, BECKII = Beck Depression Inventory, DASS21 = Depression and Anxiety Stress scale, CES-D = Centre for Epidemiologic Studies Depression scale.

[14]. Weigl (2006) was rated as moderate risk of bias in 5 of the 6 QUIPS domains and utilised a univariable screen as part of a 4-step modelling process.

**Effect of increasing BMI on the odds of a positive response to intervention.**    It was difficult to conclude whether BMI was associated with pain and function outcomes following a combined first-line intervention (Fig 4). There were only five studies that reported BMI. The effect estimates for three studies were precise and close to 1 (OR = ~0.98 to 1) suggesting no effect between BMI and a positive outcome. However, these studies reported an unadjusted

**Comorbidity by measure and cut off points**

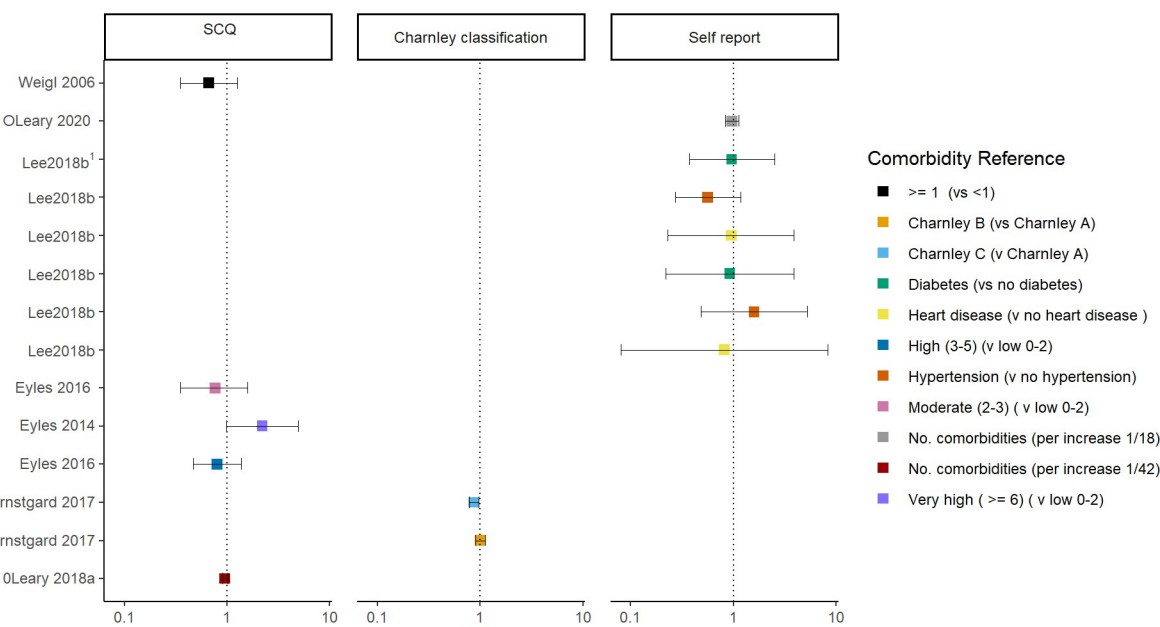

**Fig 5. Presence of comorbidity on the odds of a positive outcome following a combined first- intervention for knee osteoarthritis.**
Graph reports the log odds ratio and 95% CI. Repeated study labels by the same author represent multiple responder definitions within each study. OR > 1 for comorbidity interpreted as increased probability of being a responder with the presence of comorbidity. For a continuous predictor, we interpret the log-odds change with a one-unit change in comorbidity score. For dichotomised predictor, the OR is the probability of the comorbidity category to the reference group of being a responder. Original data from 7 studies reporting OR. Studies not included:2 reporting regression coefficient [25, 47], 2 mean differences [37, 41] and 1 unadjusted hazard ratio [36]. SCG = self-administered comorbidity questionnaire. Charnley classification = Charnley A (unilateral hip or knee OA), B (bilateral hip or knee OA), C (multiple joint sites hip and knee and presence of other disease affecting walking ability). Self-report = presence of one comorbidity or number of self-reported comorbidities.1. Lee2018b used multinomial logistical regression with multiple outcome definitions based on four group-based trajectories of WOMAC pain and function. OR > 1 indicates an increased probability of being in the lower pain, early improvement group. 2. There are six OR reported for Eyles (2016) due to multiple responder definitions and the number of comorbidities reported as low, moderate, and high.

effect estimate which makes interpreting the results difficult [10, 12, 18]. There may be greater certainty in the effect estimate in studies that present a multivariable-adjusted analysis [22].

The results of Tanaka (2021) are difficult to interpret. This study used two different responder definitions (a 5-point reduction in OKS and 50% reduction in NRS pain intensity) and reported a positive and negative OR (multidirectional) [17]. This was a small cohort study of 277 participants which had a dropout rate of close to 50%. The participant's average BMI was low (23 kgm2) in comparison to the remaining studies where the average participant BMI exceeded 28 kgm$^2$.

Four studies reported BMI continuously per one unit increase. One study dichotomised BMI and found BMI was associated with a negative outcome [29]. This study found that those who were obese (compared with normal BMI) had half the odds of reaching a self-reported minimum physical activity threshold of 150 minutes per week (OR 0.52, 95% CI 0.46–0.58).

**Presence of depression on the odds of a positive response to intervention.** It was difficult to conclude whether depression was associated with pain and function outcomes following a combined first-line intervention (Fig 4). In four studies the effect estimates were small, precise, and negative (OR between 0.9 and 1.01) which may suggest a small negative association between depression and a positive outcome [18, 21, 39, 42].

Overall, the interpretation of the 7 studies was challenging. An unadjusted effect estimate was reported in three studies [10, 12, 18] and the effect estimate was the largest for the three imprecise studies [10, 12, 14]. There was a large variation in depression measures and cut-off points. For the two studies that reported a pain outcome, the OR was precise and between 0.9 and 1.01 [21, 39]. Although both studies used a WOMAC pain MCID responder, Lee (2018) used a trajectory-based analysis assessing pain weekly over 12 weeks [21]. Depression scores were measured differently using a cut-off point (CES-D of greater than 16) and a continuous measure (per one unit increase in BECK-11score) [21, 39].

Weigl (2006) and Eyes (2016) used identical outcome measures (WOMAC-G, a transition scale and a combination of both) but the results were conflicting. Within studies the results often differed. For Eyles (2016), the OR was multidirectional (positive or negative) depending on the responder definition used [10]. Depression was associated with over three times the odds of a positive response when the response was defined as improved WOMAC-G score (OR 3.33, 95% CI 1.27–9.09) [10]. When using a self-reported transition scale (much worse or moderately worse), the OR was 0.71 (95% CI 0.36–1.52). The conflicting result might be explained by the small number of responders classified (n = 34) who were classified based on WOMAC-G responder definition [10].

**Presence of comorbidity on the odds of a positive response to intervention.** There was inconclusive evidence to determine whether comorbidity was associated with pain and function outcomes following a combined first-line intervention (Fig 5). The effect estimates were multidirectional (positive or negative) and imprecise. In addition, there was considerable variation in comorbidity outcome measures. For example, the use of self-reported presence of specific comorbidity or number of comorbidities [18, 21], a validated comorbidity measure reported continuously per one-unit increase [42] or dichotomised with cut-off points based on the number of self-reported comorbidities [10, 12, 14].

## Discussion

This systematic review is the first to evaluate prognostic factors associated with pain and function outcomes following combined first-line interventions of exercise therapy, OA education or weight loss for knee OA. The rationale of this review was to identify individual characteristics that may influence a person's response to combined first-line interventions for knee OA. A meta-analysis was not able to be performed due to study heterogeneity, instead, a narrative synthesis and data visualisation was conducted. Thirty-two studies were included in this review. We found being female was associated with 2–3 times increased odds of a positive response. Older age was associated with a lower odds of responding which was unlikely to be of clinical relevance. We could not conclude whether BMI, those with comorbidities or depression and OA severity (imaging) was associated with a positive response following combined first line interventions.

Our review found being female (compared to male) was associated with 2–3 times increased odds of a positive response to a combined first-line intervention. Although the magnitude of this effect appears large, evaluating whether this is clinically meaningful is difficult as it compares a female to male response. In addition, the use of a variety of responder definitions does not allow any evaluation of individual treatment response. Other prognostic studies that have examined sex are inconclusive or report conflicting findings. There is limited evidence that female sex is associated with symptomatic OA progression [69, 70], and there is conflicting evidence on whether being female is a predictor for future TKR [66, 71].

Older age was associated with a lower odds of responding to a combined first-line intervention. This small negative effect (less than 10% reduction in odds) is unlikely to be of clinical

relevance. Other studies that have examined age are inconclusive or report conflicting findings. Age did not appear to be associated with WOMAC score following group exercise therapy, but interpretation of the results is difficult given age was dichotomised (cut off point 65 years) [55]. Increasing age was found to be positively associated with progression to knee joint replacement following a first-line intervention of education and exercise, however the effect size was very small [30, 48]. Younger age was found to have a small association with becoming unwilling to undertake surgery following a multidisciplinary OA program [43]. The impact of increasing age on outcomes following TKR is also debated, with a 2021 systematic review concluding the evidence was inconsistent [72]. Our review suggests that age may not be a relevant factor in predicting response to first-line interventions for knee OA.

We could not conclude whether BMI was associated with a positive response following a combined first-line intervention. Few studies have evaluated BMI as a prognostic factor, and overall, the evidence of certainty was very low. Our analysis did focus on OR, however similar findings were found from studies that reported MD, HR and ß coefficient [25, 36, 37]. Dell'l-sola (2020) found increased BMI was associated with an increase in pain following the BOA intervention but concluded the difference was unlikely to be clinically important (ß coefficient 0.02, 95% CI 0.02–0.03) (S5 Table). Studies that examined obesity as a treatment moderator of exercise therapy have been inconclusive, with few high-quality trials and conflicting evidence to date [16, 73, 74]. The results of an RCT comparing non-weight bearing and weight-bearing exercise in those who are obese found no between group difference in pain and function outcomes [73]. An IPD meta-analysis of 11 RCT trials evaluating structured exercise programs for knee OA found lower BMI was associated with a small positive treatment response (OR 1.04, 95% CI 1.02–1.07) [75]. Our review suggests that BMI may not be a relevant factor in predicting a response to first-line interventions for knee OA.

We could not conclude whether the presence of comorbidity or depression was associated with a positive response following a combined first-line intervention. Few studies have specifically examined the impact of depression or comorbidity on conservative treatments for knee OA [37, 41, 76], despite much research examining predictors of musculoskeletal problems such as chronic lower back pain [77] or cross-sectional studies evaluating the association between comorbidities and clinical symptoms in those with knee OA [67, 78, 79].

Current evidence suggests that those with comorbidities and depression may respond to first-line interventions in similar ways [41, 47, 75]. A subgroup analysis of an internet-based exercise and education program found little difference between the number of comorbidities present and pain and function outcomes [47]. A large, well-designed registry-based cohort study examined the association between comorbidities and change in pain and function following the GLA:D® program. For both the primary outcome (change in 40-metre walk test) and secondary outcome (change in NRS pain), little difference was found in the adjusted mean difference in those with and without comorbidities (S5 Table). Although those with comorbidities had worse baseline scores across all outcomes, similar improvement was found in both groups following the intervention. A 2020 systematic review concluded there was insufficient evidence to determine whether comorbidity and depression moderate the effects of exercise therapy in people with hip and knee osteoarthritis [75]. This review also highlighted similar methodological limitations found in our review, such as failure to identify the moderator a priori, diversity in measurements and the use of arbitrary cut-off points [75].

We could not conclude whether OA severity (imaging) was associated with a positive response following a first-line intervention. There were four small exploratory studies that examined baseline imaging and comparing studies was challenging due to different imaging modalities, outcome measures and cut-off points for grading OA severity. Despite the studies reporting positive findings such as advanced PF OA, severe medial compartment OA and

higher KL grade being associated with a poor response, further research is required to be able to make any clear conclusions [18, 21, 31].

## Strengths and limitations

A limitation of this systematic review was the inability to pool the effect measures. A meta-analysis was not able to be performed due to each study varying on many dimensions such as the intervention components, the joint analysed, the follow-up duration, outcomes measures, the definition of a responder, prognostic factor measurement and the statistical methods. The knee joint was the focus of this review; however, some studies did report hip and knee data together. Separate reporting of the hip and knee in future primary studies is recommended as there are known differences in risk factors, prognosis, clinical presentation, and non-surgical recommendations [80].

We focussed our review on key prognostic factors identified in our protocol which included demographics (age and sex), BMI, psychological factors, and OA severity (imaging). This may be considered a limitation of this review as we missed potentially important factors. Our initial research question was broad, and a more specific focus is preferable in order to make useful conclusions [81]. It is acknowledged that some factors were not considered in detail. For example, several recent studies examine the association between pre-treatment pain sensitisation and outcomes following exercise therapy [39, 52, 61]. As further evidence emerges, a systematic review focussing specifically on factors such as pain sensitisation may be warranted.

We did not include exercise only interventions in our review. The decision to focus on any combination of land-based exercise therapy, education or weight management reflects the fact that these interventions are consistently recommended as first-line care in clinical guidelines and are commonly delivered together in clinical practice and OAMPs. The evaluation of these complex interventions does remain a challenge. Determining suitable combinations and the most appropriate outcome measures, as well as the mechanism to explain their effectiveness is yet to be determined [5] and beyond the scope of this review.

Restricting our review to cohort studies may simplify our analysis and certainty in the effect estimates. Longitudinal cohort studies may provide better prognostic estimates due to a broad inclusion criterion and may provide more generalisable findings [22]. However, cohort studies have a limited ability to separate the treatment response from the natural history of the disease and to determine causation.

Our decision to focus on one effect measure was a limitation but unavoidable due to the OR being reported in most studies. A dichotomised outcome variable should be interpreted with caution [81, 82]. Calculating the proportion of those who respond does not allow for visibility of individual variation in treatment response and may be subject to misclassification bias if the response definition is not well constructed [73]. MD, RR and HR were reported infrequently but still need to be considered when examining the evidence [25, 41].

A strength of this review was the ability to make inferences using an estimation approach [74]. There are examples in the literature of prognostic factor studies that present results based on statistical significance, a practice increasingly being discouraged [61, 75]. Our results were limited as we did rely on the primary studies' chosen model. Depending on the study's variable selection method, a factor of interest may have been excluded based on a non- significant finding. Therefore, no information about this factor is presented in the study [77]. Other aspects that contribute to difficulty in interpreting the findings of this review include whether the study reported an unadjusted effect estimate, inconsistent adjustment for other prognostic factors, or a variable selection method based on the data rather than expert opinion, evidence, or

biological plausibility [77]. This review consisted mainly of small-sample studies which may result in a larger effect measure, and potential publication bias or selective reporting [15].

### Research and clinical implications

Based on the limited findings of this review, there is no reason to expect those with comorbidities, depression, increased BMI, more advanced imaging findings and increasing age would not respond to an intervention consisting of exercise therapy, OA education and weight loss. First-line interventions should continue to be recommended to these individuals with knee osteoarthritis. Exercise therapy has been shown to be safe and effective for a broad range of conditions [83] and current evidence suggests some subgroups of people may respond equally to first-line interventions [41, 47, 74]. Additionally, clinicians need to recognise that similar improvements may occur irrespective of a higher baseline score [41].

Future research on individual characteristics associated with a response to a first-line intervention might be strengthened by access to IPD. IPD meta-analysis would allow for standardising the inclusion and exclusion criteria, consistent adjustment of prognostic factors, maintaining continuous factors on their original scale, reducing the need for arbitrary cut-off points, reducing the need for reporting unadjusted effect estimates and allowing for large data sets to be analysed [81]. Current work is being undertaken by the Joint Effort Initiative using IPD from OAMPs, aiming to identify prognostic factors associated with improvements in pain and function [84].

### Conclusion

Based on the limited findings of this review, it is recommended that clinicians continue to recommend first-line interventions consisting of exercise therapy, education, and weight loss, irrespective of sex, age, obesity, comorbidity, depression and imaging findings to people with knee OA. This review found no clear evidence to suggest factors such as age, sex, BMI, OA severity and presence of depression or comorbidities are associated with the response to a first-line intervention for knee OA. Current evidence indicates that some groups of people may respond equally to first-line interventions such as those with or without comorbidities. Future research using IPD meta-analysis may help overcome some of the challenges found in this review, allowing for standardisation of inclusion and exclusion criteria, a more consistent study methodology and evaluation of larger datasets.

### Supporting information

**S1 Fig. Follow up duration and joint measured.**
(DOCX)

**S1 Table. PRISMA 2020 checklist.**
(DOCX)

**S2 Table. Search strategies by individual databases.**
(DOCX)

**S3 Table. Risk of bias assessment using Quality in Prognostic Studies (QUIPS) for the 27 studies included in the systematic review.**
(DOCX)

**S4 Table. Effect measures and 95% confidence intervals for individual study results for prognostic factors (age, sex, BMI, depression, comorbidity, and baseline imaging).**
(DOCX)

**S5 Table. Rating of certainty and plain text interpretation of results.**
(DOCX)

**S1 File. Prospero protocol (CRD42021234398).**
(PDF)

## Acknowledgments

We would like to acknowledge the help of Murray Turner, a Health Information Specialist at the University of Canberra, in assisting JC with the search strategy design.

## Author Contributions

**Conceptualization:** Jacqui M. Couldrick, Andrew P. Woodward, Diana M. Perriman, Jennie M. Scarvell.

**Data curation:** Jacqui M. Couldrick, Andrew P. Woodward, M. Denika C. Silva, Joseph T. Lynch, Jennie M. Scarvell.

**Formal analysis:** Jacqui M. Couldrick, Andrew P. Woodward, M. Denika C. Silva, Joseph T. Lynch, Christian J. Barton, Jennie M. Scarvell.

**Investigation:** Jacqui M. Couldrick.

**Methodology:** Jacqui M. Couldrick, Andrew P. Woodward, M. Denika C. Silva, Joseph T. Lynch, Diana M. Perriman, Christian J. Barton, Jennie M. Scarvell.

**Project administration:** Jacqui M. Couldrick, Jennie M. Scarvell.

**Resources:** Jennie M. Scarvell.

**Software:** Jacqui M. Couldrick.

**Supervision:** Andrew P. Woodward, Joseph T. Lynch, Diana M. Perriman, Christian J. Barton, Jennie M. Scarvell.

**Visualization:** Jacqui M. Couldrick, Joseph T. Lynch, Jennie M. Scarvell.

**Writing – original draft:** Jacqui M. Couldrick, Andrew P. Woodward, Joseph T. Lynch, Diana M. Perriman, Christian J. Barton, Jennie M. Scarvell.

**Writing – review & editing:** Jacqui M. Couldrick, Andrew P. Woodward, M. Denika C. Silva, Joseph T. Lynch, Christian J. Barton, Jennie M. Scarvell.

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
