## [Decision Letter · Decision Letter 0]

9 Feb 2023

PONE-D-22-32794

Individual characteristics associated with outcomes following combined first-line interventions for knee osteoarthritis: a systematic review.

PLOS ONE

Dear Dr. Couldrick,

Thank you for submitting your manuscript to PLOS ONE. After careful consideration, we feel that it has merit but does not fully meet PLOS ONE’s publication criteria as it currently stands. Therefore, we invite you to submit a revised version of the manuscript that addresses the points raised during the review process.

We look forward to receiving your revised manuscript.

Kind regards,

Germain Honvo, Ph.D.

Academic Editor

PLOS ONE

Journal Requirements:

Reviewers' comments:

Reviewer's Responses to Questions

**Comments to the Author**

1. Is the manuscript technically sound, and do the data support the conclusions?

Reviewer #1: Yes

Reviewer #2: Yes

Reviewer #3: Yes

2. Has the statistical analysis been performed appropriately and rigorously? 

Reviewer #1: Yes

Reviewer #2: Yes

Reviewer #3: N/A

3. Have the authors made all data underlying the findings in their manuscript fully available?

Reviewer #1: Yes

Reviewer #2: Yes

Reviewer #3: Yes

4. Is the manuscript presented in an intelligible fashion and written in standard English?

Reviewer #1: Yes

Reviewer #2: Yes

Reviewer #3: Yes

5. Review Comments to the Author

Reviewer #1: This is a well-conducted systematic review of potential factors which can predict the response to first-line treatments (e.g. exercise-based therapy, weight-loss and patient education). Please find my comments below.

Comment 1: The title does not match the content of the manuscript, since there are a lot of factors which are not included.

Comment 2: It would be meaningful to indicate what the authors mean by “responders”.

Comment 3: The inclusion criteria are unclear, since it is broadly stated to be prognostic factor (it is properly demographics, BMI, comorbidities, psychological factors, and baseline disease severity). Studies on e.g. baseline physical activity levels, sensitization of the nerves and inflammatory mediators have been conducted but these are not included here. Also, psychological factors are reduced to “depression” but factors such as pain catastrophizing is often found to predict treatment outcomes for patients with osteoarthritis.

Comment 4: It is unclear which types of statistical analysis are “allowed” for the studies to be included in the review (e.g. is this only studies using correlation/regressions or do you allow studies that subgroup patients based on clinical outcomes/treatment outcomes?)

Comment 5: The studies have a follow-up ranging from just after completing the intervention to 12-months following the intervention and the prediction of these outcomes will be very different.

Comment 6: it is unclear why the data in figures 3-5 are not utilized for at meta-analysis.

Comment 7: please use the first section of the discussion as a summary of the study findings.

Reviewer #2: TITLE

ok

ABSTRACT

Conclusions: You can delete the last sentence.

INTRODUCTION

Could you use the recent bibliography about OA (DOI: 10.1007/s10067-018-4270-4 , DOI: 10.1097/TGR.0000000000000328 and doi: 10.1093/pm/pnab301)

Objective: could be re-written to one short sentence / to describe in brief.

METHODS.

ok

RESULT

Ok

DISCUSSION

Discussion: In the first paragraph, Could you be present the result in brief.

Tables

ok

Reviewer #3: Thank you for the opportunity to read the manuscript. The systematic review comprehensively discussed the factors such as age, sex, BMI, OA severity that are associated with the response to a first-line intervention for knee OA. The structure of the manuscript is complete and easy to read.

6. PLOS authors have the option to publish the peer review history of their article (what does this mean?). If published, this will include your full peer review and any attached files.

Reviewer #1: No

Reviewer #2: **Yes: **Jorge Hugo Villafañe

Reviewer #3: **Yes: **Donald Manlapaz

---

## [Author Response · Author response to Decision Letter 0]

3 Mar 2023

The specific reviewer and editor comments were addressed individually in the 'Response to reviewers' document that has been attached to the submission. 

All comments have been individually addressed and the manuscript has been changed accordingly.

---

## [Decision Letter · Decision Letter 1]

27 Mar 2023

Evidence for key individual characteristics associated with outcomes following combined first-line interventions for knee osteoarthritis: a systematic review.

PONE-D-22-32794R1

Dear Dr. Couldrick,

We’re pleased to inform you that your manuscript has been judged scientifically suitable for publication and will be formally accepted for publication once it meets all outstanding technical requirements.

Kind regards,

Germain Honvo, Ph.D.

Academic Editor

PLOS ONE

Reviewers' comments:

Reviewer's Responses to Questions

**Comments to the Author**

1. If the authors have adequately addressed your comments raised in a previous round of review and you feel that this manuscript is now acceptable for publication, you may indicate that here to bypass the “Comments to the Author” section, enter your conflict of interest statement in the “Confidential to Editor” section, and submit your "Accept" recommendation.

Reviewer #1: All comments have been addressed

Reviewer #2: (No Response)

Reviewer #3: All comments have been addressed

2. Is the manuscript technically sound, and do the data support the conclusions?

Reviewer #1: Yes

Reviewer #2: Yes

Reviewer #3: Yes

3. Has the statistical analysis been performed appropriately and rigorously? 

Reviewer #1: Yes

Reviewer #2: Yes

Reviewer #3: Yes

4. Have the authors made all data underlying the findings in their manuscript fully available?

Reviewer #1: Yes

Reviewer #2: Yes

Reviewer #3: Yes

5. Is the manuscript presented in an intelligible fashion and written in standard English?

Reviewer #1: Yes

Reviewer #2: Yes

Reviewer #3: Yes

6. Review Comments to the Author

Reviewer #1: I have no further comments

Reviewer #2: Thank you for an opportunity to review the article "Evidence for key individual characteristics associated with outcomes following combined first-line interventions for knee osteoarthritis: a systematic review.”. In my opinion, the authors have done a manuscript interesting and novel. So I have no more comments.

Reviewer #3: The issues raised by the reviewer from the previous manuscript were addressed by the authors. Thank you!

7. PLOS authors have the option to publish the peer review history of their article (what does this mean?). If published, this will include your full peer review and any attached files.

Reviewer #1: No

Reviewer #2: No

Reviewer #3: **Yes: **Donald G. Manlapaz, PhD, PT

---

## [Editor Report · Acceptance letter]

31 Mar 2023

PONE-D-22-32794R1 

Evidence for key individual characteristics associated with outcomes following combined first-line interventions for knee osteoarthritis: a systematic review. 

Dear Dr. Couldrick:

I'm pleased to inform you that your manuscript has been deemed suitable for publication in PLOS ONE. Congratulations! Your manuscript is now with our production department. 

Kind regards, 

on behalf of

Dr. Germain Honvo 

Academic Editor

PLOS ONE